# Breast Cancer Genomics: Primary and Most Common Metastases

**DOI:** 10.3390/cancers14133046

**Published:** 2022-06-21

**Authors:** Caroline Bennett, Caleb Carroll, Cooper Wright, Barbara Awad, Jeong Mi Park, Meagan Farmer, Elizabeth (Bryce) Brown, Alexis Heatherly, Stefanie Woodard

**Affiliations:** 1Birmingham Marnix E. Heersink School of Medicine, The University of Alabama, 1670 University Blvd, Birmingham, AL 35233, USA; cib@uab.edu (C.B.); cmcarro1@uab.edu (C.C.); cjwrig15@uab.edu (C.W.); 2Debusk College of Osteopathic Medicine, Lincoln Memorial University, 6965 Cumberland Gap Pkwy, Harrogate, TN 37752, USA; barbara.awad@lmunet.edu; 3Department of Radiology, The University of Alabama at Birmingham, 619 19th Street South, Birmingham, AL 35249, USA; jmpark@uabmc.edu; 4Department of Genetics, Marnix E. Heersink School of Medicine, The University of Alabama at Birmingham, 1670 University Blvd, Birmingham, AL 35233, USA; meaganfarmer@uabmc.edu (M.F.); acoates@uabmc.edu (A.H.); 5Laboratory Genetics Counselor, UAB Medical Genomics Laboratory, Kaul Human Genetics Building, 720 20th Street South, Suite 332, Birmingham, AL 35294, USA; ebfincher@uabmc.edu

**Keywords:** primary breast cancer, metastatic breast cancer, driver mutations, molecular subtypes, organotropism, breast cancer genomics

## Abstract

**Simple Summary:**

Breast cancer is a heterogeneous disease, and numerous associated genetic alterations have been identified. Knowledge of driver and other associated mutations has progressed significantly in recent years, secondarily to advances in gene sequencing. This manuscript provides an overview of genetic alterations in both primary and metastatic breast cancer.

**Abstract:**

Specific genomic alterations have been found in primary breast cancer involving driver mutations that result in tumorigenesis. Metastatic breast cancer, which is uncommon at the time of disease onset, variably impacts patients throughout the course of their disease. Both the molecular profiles and diverse genomic pathways vary in the development and progression of metastatic breast cancer. From the most common metastatic site (bone), to the rare sites such as orbital, gynecologic, or pancreatic metastases, different levels of gene expression indicate the potential involvement of numerous genes in the development and spread of breast cancer. Knowledge of these alterations can, not only help predict future disease, but also lead to advancement in breast cancer treatments. This review discusses the somatic landscape of breast primary and metastatic tumors.

## 1. Introduction

In 2021, breast cancer (BC) accounted for 30% of all cancers diagnosed in women and was the leading cause of cancer death in the United States among women aged 20–59 years [1]. In fact, one in eight US women will be diagnosed with invasive BC at some point in their life [1]. Although BC death rates have decreased by 41% from 1989 to 2018, its incidence continues to rise at a rate of 0.5% per year, partially due to increases in body weight and declines in fertility rate [1,2,3]. Implementation of more robust population-based screening mammography programs and the shift from film mammography to more sensitive digital mammography may have also contributed to this rise [4,5,6]. 

Although highly prevalent and heavily researched, many of the mechanisms underlying BC pathogenesis are still debated. Two models for BC pathogenesis, the sporadic clonal evolution model and the cancer stem cell model, have been proposed. In the sporadic clonal evolution model, random mutations can occur in any breast epithelial cell, and the cells that acquire advantageous genetic and epigenetic alterations are selected over time, eventually leading to tumor development. Alternatively, the cancer stem cell model argues that only a small subset of tumor cells, the stem and progenitor cells, can initiate and sustain tumor development. However, it has also been suggested that these two models may occur in concert via clonal evolution of stem cells [7]. A myriad of modifiable and non-modifiable risk factors may contribute to these changes. A few examples include obesity [8], chest radiation therapy, excessive alcohol consumption, diethylstilbestrol exposure, hormonal contraceptives, hormone replacement therapy after menopause, heritable gene mutations (such as inactivating mutations in *TP53* in Li-Fraumeni syndrome and truncating *PTEN* mutations in Cowden syndrome), and even mutations in mitochondrial DNA [9,10]. Although most BC is considered sporadic, approximately 5–10% of BC is due to these underlying hereditary causes, most often resulting from pathogenic variants or mutations in the *BRCA1* and *BRCA2* genes [11]. 

Since the discovery of the role of *BRCA1* and *BRCA2* in hereditary BC, successful germline analysis of *BRCA1*, *BRCA2*, and other BC susceptibility genes has provided the foundation for genetics’ utility in screening, cancer risk reduction, and treatment [12]. With the advent of next generation sequencing, thousands of primary BC tumors have been sequenced, expanding knowledge of BC’s genomic landscape. This new information has helped to identify previously unrecognized mutations [13]. However, differentiating driver mutations, which impart growth advantage and promote tumorigenesis, from non-pathogenic passenger mutations (Figure 1) is necessary for the development of individualized treatment [14,15]. Given the scarcity of high-frequency driver mutations and the vast amount of genomic information now available, identifying clinically significant and targetable mutations is a primary challenge [16,17].

Of those diagnosed with primary BC, 20–30% will develop distant metastasis. According to one SEER-based study, frequency of BC metastasis to the bone, lung, liver, and brain were 30–60% (bone), 21–32% (lung), 15–32% (liver), and 4–10% (brain) [18]. Moreover, the incidence of rare site BC metastasis continues to rise as a result of better imaging techniques and treatment advancements, which prolong survival [19]. Although metastasis is the leading cause of death for BC patients, BC diagnosed before distant metastasis occurs has a much better prognosis [13,20]. Metastases arise from primary cancer subclones and are more heterogenous, requiring specific treatment management and targets [13]. These statistics and knowledge underscore the importance in recognizing not only frequently implicated BC gene mutations but also lesser-known primary and metastatic BC driver mutations and their potential therapy targets. To aid in this endeavor, we seek to provide an overview and exploration of several common and novel mutations in BC, from primary to metastasis. 

This review focuses on literature concerning pathogenic variants or mutations associated with primary and metastatic BC. PubMed and Google Scholar were used as search databases, with no restrictions placed on country or publication date. In addition to original search queries, relevant articles for inclusion were identified in the references of previously found articles (backward search). In reviewing articles for inclusion, emphasis was placed on specific genes frequently or uniquely found in primary BC and certain metastatic sites. No particular gene approach was specified or emphasized. Inclusion criteria specified peer-reviewed publications, and any non-peer-reviewed studies were excluded.

### 1.1. Organotropism: Breast Cancer Molecular Subtypes

Efforts to better predict tumor behavior and improve BC treatment led to the classification of BC by molecular subtypes, which include luminal ER+ (luminal A and luminal B), human epidermal growth factor receptor 2 (HER2)-enriched, and basal-like [22,23]. Gene expression studies have shown that the different molecular subtypes vary markedly in not only prognosis but also in therapeutic targets [10]. In 1889, Stephen Paget postulated that BC metastasis was nonrandom, due to tumor cells (the seeds) having a predilection for specific organ sites (the soil) [24]. It has since been found that the different molecular subtypes demonstrate a propensity for specific organ sites [25]. Kennecke et al. also demonstrated that certain BC subtypes are associated with spread to specific organs, underscoring the critical role each “seed” plays in metastasis [26]. For example, bone metastasis is most common in luminal A and B subtypes, but the least common site in basal subtype [26]. The frequency of BC metastasis to the brain is most common in HER2-enriched (28.76%), basal-like (25.23%), and triple negative non-basal (22.06%) subtypes [26]. Liver metastasis is more common in the HER2-enriched compared to HER2-negative subtype [26]. One SEER-based study found that lung metastasis was more common in basal-like/triple-negative BCs (32.09%) compared to HR+/HER2−, HR+/HER2+, and HR−/HER+ BCs [18]. However, the factors influencing the timing and mechanisms involved in specific BC molecular subtype metastasis to the lung are not yet understood [27]. Aside from the molecular subtypes (Figure 2), the accumulation of mutations in cancer cells can alter critical genes and pathways and promote organ-specific metastasis [28]. As the genomic landscape of primary BC continues to be explored, the primary BC genetic alterations that promote organ-specific metastasis and survival in new immune and metabolic microenvironments continue to be elucidated.

### 1.2. Clinical Presentation and Detection of Breast Cancer

A palpable breast lump is the most commonly presenting symptom for BC, with one study finding that a breast lump was the presenting sign in 92.5% of the 8639 female BC patients evaluated [29]. Other signs and symptoms include nipple abnormalities (such as discharge or retraction), breast pain, skin changes, and, less commonly, back pain and weight loss [30,31]. Screen-detected BCs are associated with a strong survival advantage compared to symptomatic BC [32,33]. Even when accounting for lead-time and length-time bias, this advantage may be partly attributed to intrinsic biological differences between screen-detected and symptomatic BCs [33,34]. Screen-detected BCs are primarily of the less aggressive Luminal A subtype, whereas Luminal B, basal-like, and HER2-enriched subtypes are much less common [35]. Triple negative BCs, which comprise much of the basal-like BC molecular subtype, are more likely than other BCs to be detected symptomatically, in the interval between regular mammogram screenings [36,37]. These interval cancers tend to be more aggressive, larger in diameter, of higher stage and grade, and with a higher number of positive nodes compared to screen-detected cancers. They are also more likely to present with other unfavorable prognostic features such as hormone receptor negativity [38]. 

### 1.3. Clinical Implications: Successes of Targeted Therapy

Past and recent success of targeted molecular therapy heralds promise for the successful development and implementation of other therapies focused on specific genes, proteins, and pathways implicated in primary and metastatic BC. Prior to the approval of trastuzumab in 1998, the only available treatment for metastatic HER2+ invasive BC was traditional chemotherapy regimens [39]. Trastuzumab combined with chemotherapy has resulted in greatly improved survival rates in the historically aggressive HER2+ BC subtype [40,41]. Moreover, the α-specific PI3K inhibitor alpelisib combined with the estrogen receptor antagonist fulvestrant has demonstrated itself effective in the treatment of BC with *PIK3CA* mutations. These mutations are common in ER+/HER2- metastatic BC, with about 40% of HR+/HER2− BC patients having an activating mutation in the *PIK3CA* gene [42,43]. In a phase 3 clinical trial, patients with HR+/HER2−, *PIK3CA*-mutated BC who received alpelisib–fulvestrant demonstrated statistically significant and clinically meaningful progression-free survival, compared to those with HR+/HER2−, *PIK3CA*-mutated BC who received placebo–fulvestrant [44]. The development of alpelisib is critical, since many HR+ BCs develop endocrine resistance during treatment. The frequency of alterations in the MAPK signaling pathway (notably, loss-of-function mutations in *NF1* and gain-of-function mutations in *ERBB2*) in addition to transcription-associated alterations (for example, *MYC* amplifications and *CTCF* and *FOXA1* alterations) increase following endocrine treatment [45]. Mutation of the ligand-binding domain of Estrogen Receptor 1 (ESR1) encoding estrogen receptor α is the chief mechanism of resistance, particularly to aromatase inhibitors [46]. This recognition is important, considering that the majority of BCs are ER+ and, although many of these patients enter remission, half will relapse or progress to incurable metastatic disease. Compared to endocrine therapy alone, the CD4/6 inhibitors palbociclib, ribociclib, and abemaciclib, and the mTOR inhibitor, everolimus, show improved progression-free survival following metastasis when combined with endocrine therapy [47,48,49,50,51,52]. Additionally, utilization of the PARP inhibitors olaparib and talazoparib for germline *BRCA* mutations carriers with metastatic HER2− BC showed significant improvement in progression-free survival and health-related quality of life (although not overall survival), compared to chemotherapy alone [53,54,55]. With the success of these treatments, a precedent has been set for the discovery of other actionable and commonly shared BC drug targets [9]. Table 1 provides some of these therapies, mechanisms, targets, and advancements.

## 2. Role of Genetics and Genomics in Primary Breast Cancer

### 2.1. Overview

Current knowledge indicates that only a select number of driver genes are routinely implicated in primary BC. These include, with alteration frequencies of about 30%, *PIK3CA* and *TP53*, and, with amplification frequencies around 15%, *ERBB2*, *FGFR1*, and *CCND*. Nearly all primary BCs consist of a dominant subclone (>50% of tumor cells), commonly comprised of *TP53* and *PIK3CA* mutations [17]. In basal-like tumors, *TP53* loss of function mutation occurs at frequencies as high as 80%, rendering it a potential target of interest. Successful therapy against *HER2* amplifications has established a precedent for identifying gene targets in cancer subtypes, such as basal-like cancers, where chemotherapy is the only established medical treatment [56]. Sequencing the exons of 173 genes in 2433 primary breast tumors likewise identified *PIK3CA* and *TP53* mutations in 40.1% and 35.4% of samples, respectively. Just five other genes—*MUC16*, *AHNAK2*, *SYNE1*, *KMT2C*, and *GATA3*—were found in at least 10% of the samples. Notably, several different cancers harbor *MUC16*, *AHNAK2*, and *SYNE1* mutations, although their significance is still unknown [57]. Additionally, the mapping of all mutations in 31 tumor types to variant interpretations merged from three genetic knowledge bases showed BC to have the greatest fraction of patients who might benefit from existing investigational targeted therapies, due to frequent mutations of *AKT1*, *ERBB2*, and *PIK3CA* [58]. Important to note, however, is that many driver mutations have low allele fractions, and even uncommon driver mutations may be clinically relevant therapeutic targets [17].

### 2.2. Other Genes of Interest

Multiple copy number alterations have also been identified in HER2+ BC, including loss of *PTEN* and *INPP4B*. PI3K pathway inhibitors may prove potential therapeutic targets in BCs harboring these deletions [59]. Identified mostly in basal-like/triple-negative BC, INPP4B is a novel tumor suppressor and inhibitor of PI3K/Akt signaling and cell survival in ER+ BC and, in one study, was linked to loss of *PTEN* in BC subtypes with poor prognosis [60]. Moreover, due to high mutational frequency, *PIK3CA* activated kinase or signaling pathway inhibitors may be helpful in the treatment of luminal/ER+ cancer [61].

In a genomic analysis of 2000 breast tumors, Curtis et al. identified heterozygous and homozygous deletions in *PPP2R2A* (8p21, region 11), *MTAP* (9p21, region 15), and *MAP2K4* (17p11, region 33), which have also been observed in numerous other cancers. *PPP2R2A* is a B-regulatory subunit of the PP2A mitotic exit holoenzyme complex, and its absence has also been detected in clear cell ovarian, endometrioid, and colorectal cancers [14]. PP2A phosphatase is a serine/threonine phosphatase tumor suppressor controlling mitosis, cytokinesis, and other aspects of the cell cycle. While previous studies have reported breast tumor growth and poor outcomes associated with PP2A inhibitory protein overexpression and *PP2A* inactivation, Watt et al. demonstrated that loss of specific PP2A regulatory subunits is of likely functional significance in breast tumorigenesis [62]. Thus, activation of *PP2A* and modulation of the enzymes involved in PP2A’s suppression and inactivation could serve as potential targets for the prevention of aberrant cell cycle progression and chromosome division. *MTAP* is often co-deleted with *CDKN2A* and *CDKN2B*, and the enzyme MTAP plays a role in adenine and methionine salvage from endogenous MTA. Consequently, cells deficient in MTAP are more susceptible to de novo purine synthesis inhibitors and methionine starvation [63]. More recent studies have identified MAT2A as a viable target in *MTAP*-deleted cancers, and clinical development has begun for the MAT2A inhibitor, AG-270 [64,65]. Stephens et al. [66] identified somatic *MAP3K1* mutations in 6% of mostly ER+ BC tumors. MAP3K1 is involved in cell migration, mitosis, and apoptosis via caspase activation [66,67]. Additionally, *MAP3K1* and the more common *PIK3CA* co-occur in approximately 11% of *PIK3CA* mutant tumors and, thus, may serve as a biomarker in PI3K pathway inhibitor trials [68].

Of recent, but lesser-known, interest is the long non-coding RNA (lncRNA) homeobox *MNX1* (7q36.3), which encodes for the transcription factor motor neuron and pancreas homeobox 1 (MNX1). lncRNAs are highly sensitive regulators of cancer growth, and MNX1′s expression has been found to be upregulated in not only BC, but also in all other cancers, although its exact contribution to tumorigenesis is uncertain [69,70]. Tian et al. found elevated expression of MNX1 in patients with larger tumor size, more extensive lymph node involvement, and poorer prognosis, thus supporting *MNX1*′s role as a cancer promoter [70]. Li et al. showed that MNX1 antisense RNA 1 (MNX1-AS1) interacted with and upregulated STAT3′s phosphorylation via enhanced STAT3 and p-JAK interaction five times more in triple negative breast cancer (TNBC) versus normal breast and non-TNBC tissue. An additional in vivo experiment involving nude mice demonstrated that the silencing of MNX1-AS1 reduces tumor growth and lung metastasis, further supporting MNX1-AS1 as a novel therapeutic target in TNBC [71]. Table 2 provides a non-comprehensive overview of genes discussed in this review.

## 3. Genomics of Metastasis: Most Common Sites

### 3.1. Bone Metastasis

Bone is the most common location for BC metastases. Of the main subtypes of BC, luminal hormone positive (HR+) tumors and even more specifically, ER+ tumors, have the strongest propensity for bone metastasis (Figure 3) [72]. A study analyzing gene expression in 69 bone metastasis samples and 39 non-bone metastasis samples revealed 69 differentially expressed genes. Five genes demonstrated significantly higher expression in breast cancer bone metastasis (BCBM): *TFF1*, *TFF3*, *AGR2*, *NAT1*, and *CR1P1* [73]. The highest-ranking gene expressed, *TFF1*, produces a cysteine rich protein that is normally expressed in the gastric mucosa. The overexpression of *TFF1* in ER+ BC was later supported; however, its functional role in BCBM metastasis, if any, is still unknown [74].

Additionally, a limited retrospective study on archival tissue of 41 metastatic BCs found increased expression of the chemokine receptors CXCR4 and CCR7 [76]. CXCR4 was exclusively observed in BCBM, whereas CCR7 was also observed in other metastases, albeit expressed in a larger percentage of the bone metastasis cases. CXCR4 was noted to work in close relation to CXCL12, acting as a signal for the chemotaxis and migration of BCBM [77]. There is also evidence suggesting that an upregulation of SNAI1 is associated with BCBM [78]. SNAI1 is a zinc finger transcriptional repressor of CDH1, which encodes E-cadherin. Downregulation of E-cadherin is necessary for the dissemination and invasion of cancer cells, loss of epithelial differentiation, and acquisition of a mesenchymal phenotype, which might augment BC metastasis into the bone [79]. 

A multigenetic study in mice, utilizing the parental BC cell line MDA-MB-231, was conducted to analyze BCBM’s genetic associations. Subpopulations of the parental cells were organized based upon their preference of metastasis location. Compared to other sites, the subgroups with first metastasis to the bone showed a substantial increase in IL11, CTGF, CXCR4, and MMP-1 expression. When IL11 was expressed alone, it did not show highly metastatic effects, but when in combination with osteopontin, it was highly metastatic to bone. Osteopontin, however, was found in many highly metastatic populations and was not specific to bone. Similarly, with the remaining genes, the individual expression was not overtly metastatic, but instead, the summation of three or four of the genes resulting in a highly bone specified metastatic tumor was found. Although this work was completed in a mouse model, it aids in demonstrating the functional pro-metastatic effect of these specific genes [77].

Notably, a study of 389 primary BC tumors found no significant somatic mutation associations in BCBM for their cohort of samples. It should be stated, however, that the study analyzed only 46–50 cancer related genes in each tumor sample. Although no genes were found to be specifically associated with bone metastasis, increased expression of *TP53*, *PIK3CA*, and *AKT*, which have been previously implicated in many other BC metastases, was observed [80].

### 3.2. Lung Metastasis

As BC lung metastasis (BCLuM) is typically symptomatic only once the lungs are overtaken with secondary tumors, identification of its cellular and molecular processes is critical for treatment development [81]. The basal-like subtype of BC has a propensity for BCLuM [82], and another study of 2933 BC patients found that 75.8% of BCs that gave first rise to lung metastasis expressed either HER2 or EGFR [78]. Minn et al. identified a set of genes that not only promote, but are also clinically correlated with BCLuM. They include *MMP1*, *MMP2*, *CXCL1*, *PTGS2*, *ID1*, *VCAM1*, *EREG*, *SPARC*, and *IL13RA2*. Many of these genes encode for metastasis-promoting extracellular products such as growth and survival factors (the HER/ErbB receptor ligand epiregulin), chemokines (CXCL1), cell adhesion receptors (ROBO1), and extracellular proteases (MMP1). Others, such as ID1 and COX2, encode for transcriptional regulators and intracellular enzymes, respectively [83].

Overexpressed in ER+ BC cells, *AGR2* may also promote BCLuM via the (de)regulation of tumor cell adhesion and spread [81]. AGR2′s potential role in tumorigenesis can be attributed to its likely function in protein folding and endoplasmic reticulum-assisted degradation of proteins [84]. Thus, *AGR2* may facilitate tumor resistance against proteotoxic stress, preventing tumor cell death. With therapies targeting inhibition of AGR2, tumor cells are more susceptible to proteotoxic stress, which helps prevent tumor metastasis and slow its spread [85]. 

Additionally, KLF5 is a transcription factor that is expressed in high-grade ER− tumors, such as basal-like BC [86]. It promotes overall BC cell development, survival, spread, and overall tumor growth [87] and has been found to promote BC cell proliferation through its target genes *FGF-BP*, *mPGES1*, p27, and the tumor necrosis factor-α induced gene, *TNFAIP2* [88,89,90,91]. Specifically, TNFAIP2 interacts with the two small GTPases Rac1 and Cdc42, increasing motility via their activities, to alter actin cytoskeleton and cell morphology. Therefore, the proliferation, migration, and invasion of triple negative BC can be stimulated by KLF5-upregulation of TNFAIP2 [88]. KLF5′s function is also stabilized by the deubiquitinase, BAP1. In fact, BAP1 helps promote BCLuM via KLF5 stabilization [87].

Only a small fraction of cancer cells from the primary tumor successfully metastasizes. What allows these cancer cells to proliferate at a particular site is dependent on that organ’s microenvironment [92]. In an in vivo mice study, Salvador et al. demonstrated that *Loxl2* facilitates the premetastatic niche, thereby promoting dedifferentiation and tumor invasion. In contrast, it was demonstrated that *Loxl2* deletion in the primary breast tumor led to a marked decrease in lung metastatic burden, whereas its overexpression produced the opposite effect [93].

### 3.3. Liver Metastasis

Breast cancer liver metastasis (BCLM) is a complex process, involving specific gene mutations with few known therapeutic options, which often leads to poor outcomes. Liver metastasis is reported in 15% of newly diagnosed BC patients and is a complex multistep process, involving signaling pathways [94]. The MAPK, NFκB, and VEGF signaling pathways are mechanisms highlighted in regulating BCLM. According to one study [95], the prognosis following liver metastasis has the second-worst outcome after breast cancer brain metastasis (BCBrM), and about half of patients with metastatic BC eventually develop liver metastasis. BCLM has low expression of immune genes in comparison to other organ sites. BCLM has a very poor prognosis and has a survival time of four to eight months if left untreated, even though new therapies in the last decade have resulted in a yearly 1–2% decrease in mortality rates [94]. The many signaling pathways and genetic mutations involved in BCLM contribute to the poor prognosis associated with the disease. 

A study conducted by Chen et al. [94] identified MAPK, NFκB, and VEGF as the three most critical molecular pathways in the regulation of BCLM. These three pathways may be responsible for the distinct pathology seen in liver metastasis, due to the high observed gene count in BCLM genomes. The MAPK cascade is the most important mitogenic pathway to human cancer pathogenesis. Activation of MAPK can lead to downstream changes in both protein expression and activity, because of its role in growth factor mediation. In contrast, the NFκB pathway plays a role in apoptosis, allowing cancer cells to initiate cell death in healthy cells. Finally, the VEGF pathway enables cancer cells to induce new blood vessel formation and growth, which allows the tumor to expand and travel throughout the body, leading to metastasis. 

Mutations in driver genes *ESR1*, *AKT1*, *ERBB2*, *FGFR4*, and the MS APOBEC cytidine deaminases negatively alter cellular mechanisms. Defective DNA mismatch repair further contributes to the progression of liver metastasis. *ESR1* has been discovered as the most mutually exclusive mutant gene pair in liver metastasis [96]. *ESR1* is an estrogen receptor protein coding gene that is a biological indicator of tumor status in BC. Most of these mutations occur in BC cells during metastasis to the liver. 

*PPFIA1*, which has a key oncogenic role in BC, was discovered to also contribute to metastatic relapse, due to its upregulation in BCLM [97]. *PPFIA1* is a protein coding gene that is important in regulating cell migration and invasion. Expression of *PPFIA1* is significantly higher in liver metastatic breast tumors compared to primary tumors. 

Rhodes et al. [98] determined that activation of LKB1 signaling represents the possibility of developing new therapeutics, particularly for patients exhibiting basal-like BC or triple-negative breast disease with low endogenous LKB1 expression. The lack of frequent LKB1 mutations in sporadic BC further supports the possibility for successful therapeutic intervention of normal LKB1 signaling. LKB1 is a kinase that acts as a blockade upstream to many oncogenic pathways, and LKB1 expression could regulate the invasive and metastatic properties of the basal-like BC subtype. Therefore, targeting LKB1 expression by increasing the kinase mechanisms in patients with BCLM could be a possible treatment target, as it can suppress invasion and metastasis of certain BC types. 

### 3.4. Brain Metastasis

The brain is the third most common site for BC metastasis, behind bone and liver [28]. Epidemiologic studies have found that 10–16% of BC patients have brain metastases, while large autopsy studies have suggested frequencies as high as 18–30% [99]. Brain metastasis is typically diagnosed within 2–3 years of initial BC diagnosis, and it usually develops after metastasis to the bone, lung, and/or liver [100]. The median survival for patients with BCBrM is 13 months, and fewer than 2% of patients with BCBrM survive past 2 years [99]. 

Each of the PAM50-based BC subtypes exhibits a unique tropism for different metastasis locations, as exhibited by a study [26] conducted in 2010 on the metastatic behavior of these subtypes. The frequency of BCBrM from each PAM50 subtype examined in this study were: Luminal A = 7.6% (ER+ and/or PR+, Ki-67 < 14%); Luminal B = 10.8% (ER+ and/or PR+, and Ki-67 ≥ 14%); Luminal/HER2 = 15.45% (HER2+, ER+ and/or PR+); HER2 enriched = 28.76% (HER2+, ER−, and PR−); Basal-like = 25.23% (HER2−, ER−, and PR−, EGFR+ and/or CK5/6+); Triple Negative non-basal = 22.06% (HER2−, ER−, PR−, EGFR−, and CK5/6−). The more common HER2+ and Triple Negative are illustrated in Figure 4 [75].

Several genes and deeper genomic similarities have been identified in BCs with subsequent brain metastases. For instance, a 2004 study [101] found that cells from BCs selective for brain metastasis produced higher levels of the angiogenic factors VEGF-A and IL-8 in vitro compared to the parent line of BC cells. A 2014 study [99] found that BCs with brain metastasis exhibited large chromosomal gains in 1q, 5p, 8q, 11q, and 20q, and chromosomal deletions in 8p, 17p, 21p, and Xq. 

Several genes have exhibited overexpression, such as *ATAD2*, *DERL1*, and *NEK2A*. *ATAD2* is believed to be a transcription coactivator of *ESR1*, enabling the expression of many estradiol target genes, and may also be required for histone hyperacetylation [102]. Moreover, ATAD2 is a known cofactor for the oncogene *MYC*, which is associated with poor BC outcomes [103]. *DERL1* encodes a member of the derlin family of proteins, which are responsible for the endoplasmic reticulum (ER)-associated translocation of misfolded proteins for proteasomal degradation. BC cells have shown increased DERL1 expression during ER-stress (which is associated with solid tumor progression), while knockout of *DERL1* leads to decreased cancer cell development [104]. *NEK2A* encodes a serine/threonine kinase involved in mitotic regulation, which has been found to contribute to the growth potential of ductal carcinoma in situ and invasive ductal carcinoma. Additionally, NEK2A expression is correlated to a higher histological grade and lymph node metastasis [105].

The *ATM*, *CRYAB*, and *HSPB2* genes are commonly deleted and/or under-expressed in BCBrM. The *CRYAB* and *HSPB2* genes are both members of the multi-gene small heat shock protein family located on 11q23, and their specific biological roles are unclear. However, 11q23 is believed to be a tumor suppressor for many solid tumors, including breast, cervical, ovarian, gastric, bladder carcinoma, and melanoma [106]. The *ATM* gene encodes the ATM serine/threonine kinase, which has a major role in DNA repair, and the absence of ATM expression leads to genomic instability and cancer predisposition [107].

Certain cellular pathways were also found to be enriched in BC cells with brain metastases. The “IL-8 signaling” pathway demonstrated enrichment, mainly as a result of hypermethylation and downregulation of many genes, including *ANGPT1*, *KDR*, *ITGAM*, *PIK3CG*, and *TEK*. *BRAF* and *BCL2*, in contrast, were hypomethylated and overexpressed in this pathway. Similarly, the cellular pathways of “hepatic fibrosis/hepatic stellate cell activation signaling” and “thyroid hormone metabolism signaling” were also frequently enriched. Genes involved in cell cycle progression and the G2/M transition pathway have also been found to be enriched, such as *AURKA*, *AURKB*, and *FOXM* [99].

DNA methylation has also shown to play a role in BCBrM. Hypermethylation and downregulation of the *PENK*, *EDN3*, *RELN*, and *ITGAM* genes appear to cause defects in cell migration and adhesion in BCBrM, while hypomethylation and subsequent upregulation of the *KRT8* gene increased cell adhesion and permeability in cancer cells. Another study [108] from 2004 found the *HIN1* and *RARB* genes to be hypermethylated in brain metastases compared to primary BCs. These data suggest that demethylating agents may have therapeutic significance for BCBrM patients. 

## 4. Genomics of Breast Cancer Metastasis: Selected Rare Sites

### 4.1. Orbital Metastasis

Orbital metastases (OM) are an infrequent occurrence, comprising anywhere from 1 to 13% of malignant orbital tumors [109,110,111]. Although the breast is the most common primary site of OM, small studies provide limited insight as to its true incidence, with reports ranging anywhere from 29 to 53% [112,113]. One systematic meta-analysis [114] of 72 orbital metastatic tumors found that infiltrating lobular breast cancer (ILBC) was five-times more likely to be implicated in OM than invasive ductal BC, although it represents just 10–15% of mammary carcinomas overall. ILBC has been associated with loss of E-cadherin, P-cadherin, HER2, EGFR, and p53 expression, and estrogen promotes its growth [115]. In a study of eight OMs from BC tumors, seven exhibited MGBN, GATA3, and BCL2 positivity, as well as ERBB2 negativity. A concordant *PIK3CA* activating oncogene mutation was found in both the primary tumor and OM of one patient, with findings consistent with ILBC [114]. BC ocular tropism may be linked to the expression of estrogen receptors in normal conjunctiva, tear glands, and tarsal conjunctiva [116,117]. Additionally, periorbital fat produces the steroid hormones required for lacrimation, potentially explaining hormone-sensitive tumors’ affinity for the orbit [117]. This suggestion is supported in another study [118], where 16/20 orbital metastases were confirmed as ER+.

### 4.2. Gynecologic Metastasis

Gynecologic metastases are uncommon across all cancer types. Due to proximity, colorectal cancer most commonly metastasizes to this area, followed by BC, indicating BC’s spread is likely targeted. As a possible consequence of hormone signaling, patients with BC gynecologic metastases present at a younger age (46–54 years) with ER+ and HER2− ILBC [119]. Kutasovic et al. found recurrent gene amplifications at *FGFR1* and *CCND1* that coincided with therapy resistance in ER+ BC metastasized to gynecologic sites. Additionally, many of the metastatic tumors harbored these mutations solely in the metastasis, indicating these mutations were either present in an unsampled portion of the primary tumor or later acquired as a means for dissemination and/or survival [119]. FGFR1 signaling through the MAPK and PI3K pathways has been specifically implicated in BC growth, survival, and metastasis. Moreover, 7.5–17% of all BCs and 16–27% of luminal-B type tumors possess a *FGFR1* mutation. *FGFR1* amplifications often demonstrate therapy resistance [120]. However, several studies have shown promise in the utility and practicality of targeting *FGFR1* [121]. 

### 4.3. Pancreatic Metastasis

Comprising just 2–5% of pancreatic tumors overall, metastases to the pancreas is a rare event [122]. Of these, the breast is the source of only 3–5% of these metastases [123]. Owing to this, genetic analyses of BC metastases to the pancreas are limited. One study [123] investigating biomarkers for BC to pancreas metastasis identified a *ERBB2 I767M* mutation within the pancreatic tumor consistent with BC origin. This gene mutation’s functional significance is undetermined. Although one study [124] found no distinction in cell proliferative ability or viability between wild-type and *I767M HER2*, two other studies [125,126] found evidence for *I767M* as a gain of function mutation, leading to enhanced HER2 kinase activity and AKT phosphorylation. 

## 5. Paired Primary and Metastatic Breast Cancer

Although there has been extensive investigation of the genes involved in both primary and metastatic BC, the exact events leading to dissemination of disease have not been completely elucidated. However, progress has been made in understanding the links between certain signaling pathways and the trigger for metastasis. Paul et al. studied this link by executing whole-exome and shallow whole-genome sequencing. The authors found specific genes that were not involved in primary BC but found in certain metastases. Seven of the genes included *MYLK*, *PEAK1*, *ESR1*, *PALB2*, *XIRP2*, *EVC2*, and *SLC2A4RG*. In addition, regions including *STK11*, *CDKN2A/B* loss and *PTK6*, * PAQR8* gain were also involved. The specific pathways implicated in metastases involved mTOR, CDK/RB, WNT, HKMT, cAMP/PKA, and focal adhesion. Importantly, these pathways can potentially have implications in the treatment of metastatic BC by the development of said pathway inhibitors [127]. PARP inhibitors, for instance (as previously discussed), are currently being used in clinical trials, to treat metastatic BC.

## 6. Conclusions

Both shared and unique BC mutations occur across its genomic landscape, from primary to metastasis. There are commonly implicated driver mutations in primary BC, and the wide array of genes involved in metastatic BC are becoming better understood. Furthermore, an important step to bring precision medicine into practical use is to assess non-pathogenic passenger mutations, to identify therapeutically targetable and, ideally, commonly shared driver mutations. However, even infrequent mutations that reside in known targetable pathways may hold functional therapeutic significance. While some mutations are conserved in both primary and metastatic tumors, others may be specific to certain metastatic sites and serve as genetic biomarkers and targets in both the prevention and treatment of BC metastasis. As research continues, treatment development and the understanding of BC tumorigenesis will move closer to precision. 

## Figures and Tables

**Figure 1 cancers-14-03046-f001:**
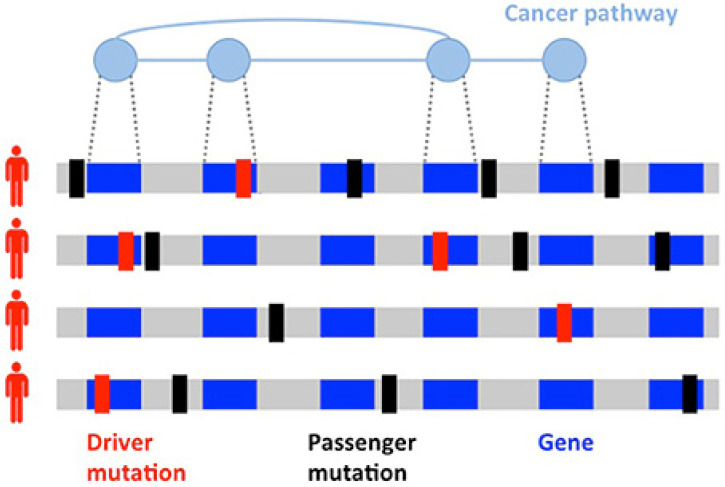
Heterogeneity of cancers. Driver (red) and passenger (black) mutations in the development of a tumor. Driver mutations are much less common than passenger mutations and involve genes implicated in cancer pathways. Figure reproduced with permission by Vandin [21]. http://creativecommons.org/licenses/by/4.0/, accessed on 2 May 2022.

**Figure 2 cancers-14-03046-f002:**
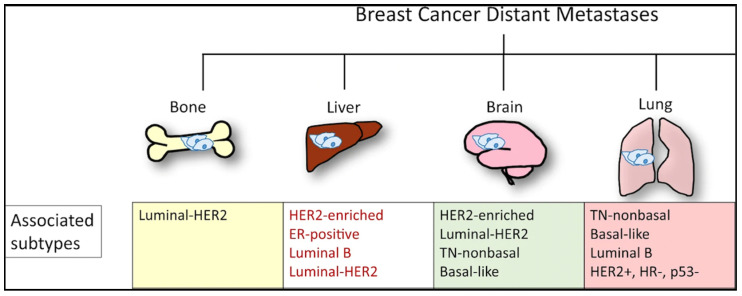
Molecular subtypes of BC and sites of the most commonly reported metastases. From Chen W, Hoffman AD, Liu H, Liu X. Organotropism: new insights into molecular mechanisms of breast cancer metastasis. *NPJ Precis. Oncol.* 2018, 2, 4. DOI: 10.1038/s41698-018-0047-0 [28]. http://creativecommons.org/licenses/by/4.0/, accessed on 2 May 2022.

**Figure 3 cancers-14-03046-f003:**
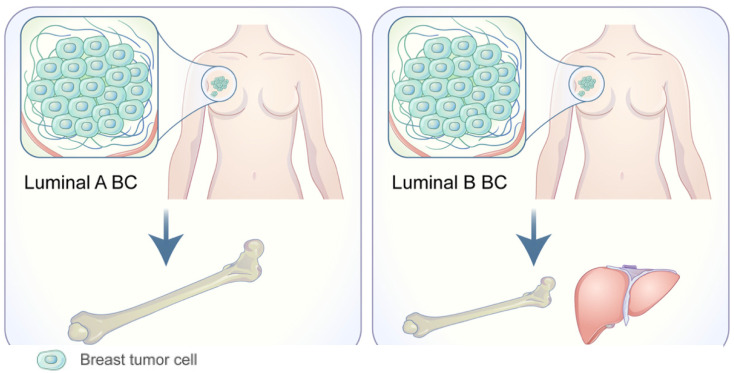
Adapted from Wang C, Xu K, Wang R. Heterogeneity of BCSCs contributes to the metastatic organotropism of breast cancer. *J. Exp. Clin. Cancer Res.* 2021, 40, 370 (2021). http://doi.org/10.1186/s13046-021-02164-6 [25,75]. http://creativecommons.org/licenses/by/4.0/, accessed 2 May 2022.

**Figure 4 cancers-14-03046-f004:**
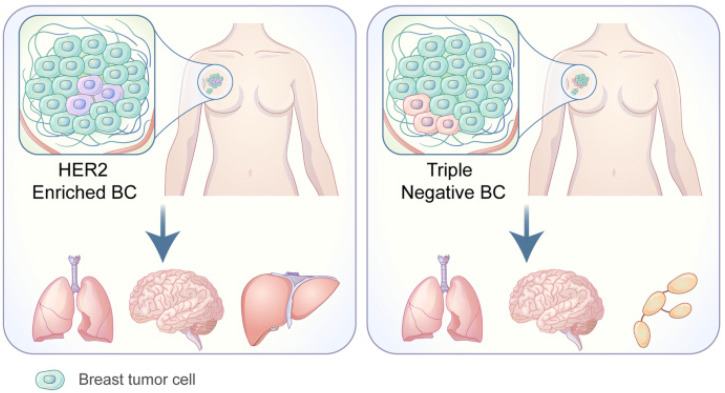
Adapted from Wang C, Xu K, Wang R. Heterogeneity of BCSCs contributes to the metastatic organotropism of breast cancer. *J. Exp. Clin. Cancer Res.* 2021, 40, 370 (2021). http://doi.org/10.1186/s13046-021-02164-6 [25,75]. http://creativecommons.org/licenses/by/4.0/, accessed 2 May 2022.

**Table 1 cancers-14-03046-t001:** Select successful targeted BC therapies, mechanisms, targets, and treatment advancements. References for this table include those provided in the literature review.

Therapy	Mechanism	BC Molecular Subtype Target	Advancement
Trastuzumab	HER2 receptor inhibitor	HER2+	Improved survival
Alpelisib	α-specific PI3K inhibitor	HR+/HER2− metastatic BCs with *PIK3CA* mutation	IPFS, when combined with fulvestrant
Palbociclib	CD4/6 inhibitor	HR+/HER2−	IPFS following metastasis, when combined with endocrine therapy
Ribociclib	CD4/6 inhibitor	HR+/HER2−	IPFS following metastasis, when combined with endocrine therapy
Abemaciclib	CD4/6 inhibitor	HR+/HER2−	IPFS following metastasis, when combined with endocrine therapy
Everolimus	mTOR inhibitor	HR+/HER2− metastatic BC	IPFS following metastasis, when combined with endocrine therapy
Olaparib	PARP inhibitor	HER2− metastatic BCs in germline *BRCA* mutation carriers	IPFS and health-related quality of life
Talazoparib	PARP inhibitor	HER2− metastatic BC (in germline *BRCA* mutation carriers)	IPFS and health-related quality of life

IPFS = improved progression-free survival.

**Table 2 cancers-14-03046-t002:** Several of the genes linked to primary and metastatic breast cancer discussed in this review. Those presented below are not intended to reflect a comprehensive list, but rather the ones highlighted in this literature review and prevalent throughout literature. References for this table include those provided in the review. Bold indicates genes that our research indicated were commonly involved in more than one site (*ERBB2*, *AKT1*), noting that this is a very limited overview.

Primary Breast Cancer	Bone	Lung	Liver	Brain
*PIK3CA* *TP53* * **ERBB2** * *FGFR1* *CCND* *MUC16* *AHNAK2* *SYNE1* *KMT2C* *GATA3* * **AKT1** * *PTEN* *INPP4B* *PPP2R2A* *MTAP* *MAP2K4* *MNX1*	*TFF1* *TFF3* *AGR2* *NAT1* *CR1P1* CXCR4 CCR7 SNAI1 IL11 CTGF CXCR4 MMP-1	*MMP1* *MMP2* *CXCL1* *PTGS2* *ID1* *VCAM1 EREG* *SPARC* *IL13RA2* *AGR2* KLF5 *Loxl2*	MAPK NFκB VEGF *ESR1* ***AKT1*** ***ERBB2*** *FGFR4* MS APOBEC cytidine deaminases *PPFIA1*	PAM50 VEGF-A IL-8 *ATAD2 DERL1* *NEK2A* *ATM* *CRYAB* *HSPB2* *ANGPT1* *KDR* *ITGAM* *PIK3CG* *TEK* *BRAF* *BCL2* *AURKA* *AURKB* *FOXM1*

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
