# Peer review of "Breast Cancer Genomics: Primary and Most Common Metastases"

_cancers, 2022, doi:10.3390/cancers14133046_

Round 1

Reviewer 1 Report

Dear Authors:

The manuscript is well written and organized. It summarized the metastasis of breast cancer clearly. Strongly suggest for publishment.

Best,

Author Response

Thank you for these remarks. We have thoroughly reviewed for any grammatical errors and feel these have been addressed. As requested, the English composition and grammar has been reviewed and approved by those whose native language is English.

Reviewer 2 Report

Dear Authors,

The article is interesting. The approach is very complex.

Here are my observations/suggestions/comments:

1.      Abstract – please provide at least one paragraph on gene approach based on your research of literature  (particular, not general)

2.      Introduction – BC incidence seems to increase nowadays also due to modern detection methods and screening protocols in some geographic areas

3.      Section 1.1. – once you introduced an abbreviation, please use it every time (e.g. BC)

4.      Clinical implications – I suggest first some clinical aspects on presentation  followed by therapy section

5.      There is no reference here

When IL11 was expressed alone, it did not show highly metastatic effects, but when in combination with osteopontin, it was highly metastatic to bone. Osteopontin, however, was found in many highly metastatic populations and was not specific to bone. Similarly, with the remaining genes, the indi-vidual expression was not overtly metastatic, but instead the summation of three or four of the genes resulting in a highly bone specified metastatic tumor. Although this work was completed in a mouse model, it aids in demonstrating the functional prometastatic effect of these specific genes.”

6.      Here = the reference should be placed at the end of the paragraph

“Notably, a study of 389 primary BC tumors found no significant somatic mutation associations in BCBM for their cohort of samples67. It should be stated, however, that the study analyzed only 46-50 cancer related genes in each tumor sample. Although no genes were found to be specifically associated with bone metastasis, increased expression of TP53, PIK3CA, and AKT, which have been previously implicated in many other BC me-tastases, was observed.”

7.      Please re-phrase this legend. This will be an article or paper or review of literature, not a “manuscript”.  “Our authors”?  I suggest “we found” or “our research  indicated..”

8.      Table 1. Several of the genes linked to primary and metastatic breast cancer discussed in this man-uscript. Those presented below are not intended to reflect a comprehensive list, but rather the ones highlighted by our authors and prevalent throughout literature. References for this table include those provided for the manuscript. Bold indicates genes that our authors found commonly involved in more than one site (ERBB2, AKT1), noting that this is a very limited overview.

9.      Figure 1-4 . Do you mean you have the permission to reproduce it or adjust it? I looked to the primary source and they are similar to me.

Author Response

  1. Abstract – please provide at least one paragraph on gene approach based on your research of literature  (particular, not general)
  • Thank you for this suggestion. We added information on this at the end of the Introduction (page 3 paragraph 1). We were focused on choosing quality, peer-reviewed papers to gather our data and wanted to choose literature supporting organization by an organ-based approach rather than into the specific gene testing approach. This is now briefly stated in order to clarify the data collection process. We chose to add this to the end of the Introduction rather than the abstract since it could be distracting given that it was not the focus of our data presentation.
  1. Introduction – BC incidence seems to increase nowadays also due to modern detection methods and screening protocols in some geographic areas 
  • We have acknowledged this in the 2. Clinical Presentation and Detection of Breast Cancer section and appreciate this comment.
  1. Section 1.1. – once you introduced an abbreviation, please use it every time (e.g. BC) 
  • Thank you for alerting us of this error. We have addressed this throughout the paper.
  1. Clinical implications – I suggest first some clinical aspects on presentation followed by therapy section
  • We have added a section on this as requested, 2. Clinical Presentation and Detection of Breast Cancer. We have discussed both the clinical presentation of patients as well as information on detection.
  1. There is no reference here

“When IL11 was expressed alone, it did not show highly metastatic effects, but when in combination with osteopontin, it was highly metastatic to bone. Osteopontin, however, was found in many highly metastatic populations and was not specific to bone. Similarly, with the remaining genes, the indi-vidual expression was not overtly metastatic, but instead the summation of three or four of the genes resulting in a highly bone specified metastatic tumor. Although this work was completed in a mouse model, it aids in demonstrating the functional prometastatic effect of these specific genes.”

  • We apologize for this error. We have corrected this.
  1. Here = the reference should be placed at the end of the paragraph 

“Notably, a study of 389 primary BC tumors found no significant somatic mutation associations in BCBM for their cohort of samples67. It should be stated, however, that the study analyzed only 46-50 cancer related genes in each tumor sample. Although no genes were found to be specifically associated with bone metastasis, increased expression of TP53, PIK3CA, and AKT, which have been previously implicated in many other BC me-tastases, was observed.”

  • We apologize for this error. We have corrected this.
  1. Please re-phrase this legend. This will be an article or paper or review of literature, not a “manuscript”.  “Our authors”?  I suggest “we found” or “our research  indicated..”Table 1. Several of the genes linked to primary and metastatic breast cancer discussed in this manuscript. Those presented below are not intended to reflect a comprehensive list, but rather the ones highlighted by our authors and prevalent throughout literature. References for this table include those provided for the manuscript. Bold indicates genes that our authors found commonly involved in more than one site (ERBB2, AKT1), noting that this is a very limited overview.
  • Thank you for this suggestion. We have edited this as requested.
  1. Figure 1-4 . Do you mean you have the permission to reproduce it or adjust it? I looked to the primary source and they are similar to me. 
  • All images are CC-BY 4.0 and listed as free use. Directly quoted from the ‘Rights and Permissions’ for all images: Creative Commons Attribution 4.0 International License, which permits use, sharing, adaptation, distribution and reproduction in any medium or format, as long as you give appropriate credit to the original author(s) and the source, provide a link to the Creative Commons license, and indicate if changes were made. The images or other third party material in this article are included in the article’s Creative Commons license, unless indicated otherwise in a credit line to the material.”
    • No credit line indicated any restrictions in use. We cited each article appropriately, giving credit and noting if adaptations were made. We included the link to the CC license as required.

Reviewer 3 Report

Reviewer comments:

Comments to the Author

This review is focused on breast cancer that is a heterogeneous disease in terms of its association with many genetic alterations. Day by day, advancement in cancer research explores the driver and the causative factors such as associated mutations leading to genetic alterations in both primary and metastatic breast cancer.

The review is for the most part well written with substantial discussion of the literature. The review organization is impressive, and the figures provided were comprehensive. The references are appropriate and timely.

Minor criticisms

• Please elaborate the section “Clinical Implications: Successes of Targeted Therapy” or compile in a table all the studies advancement.

• Please undergo a thorough check of the manuscript for typographical and grammatical errors.

Author Response

  1. Please elaborate the section “Clinical Implications: Successes of Targeted Therapy” or compile in a table all the studies advancement. 
  • We have added a table summarizing the data presented in this section. The references are from the section (as noted in the table legend).
  1. Please undergo a thorough check of the manuscript for typographical and grammatical errors.
  • Thank you for this suggestion. We have thoroughly reviewed the manuscript for grammatical errors and typographical mistakes.

This manuscript is a resubmission of an earlier submission. The following is a list of the peer review reports and author responses from that submission.

Round 1

Reviewer 1 Report

This review explores the genomic landscape of breast cancer with a focus on genetic alterations identified in primary and metastatic tissue. 

  1. I could not find the methods section that clarifies how articles were selected for the review. What gene panels were used to identify the diff alterations for various studies? Did any study explore paired primary and mets ? What key words were used?
  2. The abstract will need major revisions. More clarity on what the review is about, significance, value addition etc. Rephrase " genetic associations " to exploring the somatic landscape of breast primary and metastatic tumors
  3. Table 1 would need references. What are the shared alterations between primary and across metastatic sites? were any statistically significant differences observed?
  4. Clinical implications of alterations are not discussed? What alterations are therapeutically actionable today in breast cancer?- such as PIK3CA, ESR1, BRCA- olaparib etc
  5. the authors discuss breast cancer as a whole, however, its essentially treated clinically as 3 separate diseases- triple neg, ER/PR and Her2. Biologically they are all different.
  6. Gene expression data is very tricky to interpret. How was overexpression determined? What is the clinical significance? What is the statistical significance?
  7. Conclusions will need major revisions. While the authors discuss therapeutic relevance, this was not discussed in earlier sections while discussing the various molecular variants identified in primary and metastatic tissue.

Reviewer 2 Report

Dear Authors:

The manuscript by Bennett et al has provided an overview of genetic associations in both primary and metastatic breast cancer. I have just a few suggestions

1.Please add more scheme or figure about background, therapy or metastasis mechanisms

2.Please add more background information about breast cancer in introduction and therapy against HER2. (please cite:  1. Chen et al. Semin Cancer Biol. 2020 Oct 6:S1044-579X(20)30203-0. doi: 10.1016/j.semcancer.2020.09.012. 2. https://www.frontiersin.org/articles/10.3389/fonc.2022.820968/abstract)

Best,